

# Adversarial example defense based on image reconstruction

Yu(AUST) Zhang, Huan Xu, Chengfei Pei and Gaoming Yang

School of Computer Science and Engineering, Anhui University of Science and Technology, Huainan, Anhui, China

## ABSTRACT

The rapid development of deep neural networks (DNN) has promoted the widespread application of image recognition, natural language processing, and autonomous driving. However, DNN is vulnerable to adversarial examples, such as an input sample with imperceptible perturbation which can easily invalidate the DNN and even deliberately modify the classification results. Therefore, this article proposes a preprocessing defense framework based on image compression reconstruction to achieve adversarial example defense. Firstly, the defense framework performs pixel depth compression on the input image based on the sensitivity of the adversarial example to eliminate adversarial perturbations. Secondly, we use the super-resolution image reconstruction network to restore the image quality and then map the adversarial example to the clean image. Therefore, there is no need to modify the network structure of the classifier model, and it can be easily combined with other defense methods. Finally, we evaluate the algorithm with MNIST, Fashion-MNIST, and CIFAR-10 datasets; the experimental results show that our approach outperforms current techniques in the task of defending against adversarial example attacks.

## INTRODUCTION

Deep neural networks have been widely used in computer vision, natural language processing, speech recognition, and other fields (*Karen & Andrew, 2015*). However, the adversarial example proposed by *Szegedy et al. (2013)*, as shown in Fig. 1, can easily deceive the neural network by adding a minor perturbation to the ordinary image, *i.e.*, the deep convolutional neural network will continuously amplify this perturbation, which is sufficient to drive the model to make high confidence incorrect predictions without being detected by the human eye. As a result, the adversarial example has a minor perturbation than the normal noise. However, it brings more significant obstacles to practical applications. Researchers usually input the pictures directly into the neural network for the computer classification test when training the classifier model to solve this problem. *Kurakin, Goodfellow & Bengio (2016)* found that a significant fraction of adversarial images crafted using the original network are misclassified even when fed to the classifier through the camera. Nowadays, the research and implementation of autonomous driving (*Deng et al., 2020*) and person detection (*Thys, Ranst & Goedemé, 2019*) rely heavily

Corresponding author
Yu(AUST) Zhang,
yuzhang@aust.edu.cn

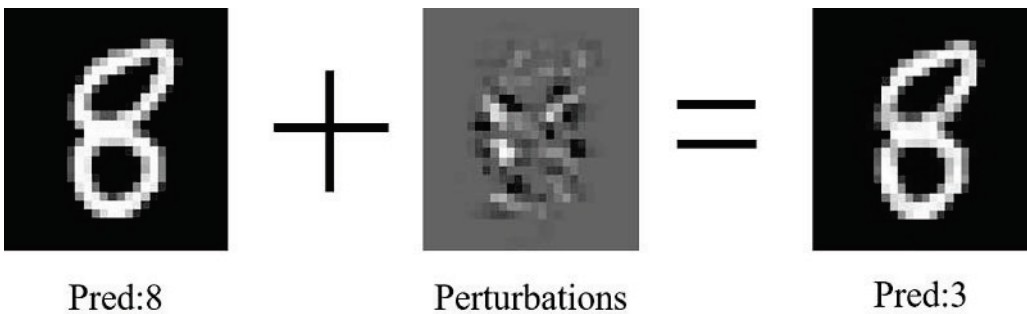

Pred:8      Perturbations      Pred:3

**Figure 1** **The generation process of adversarial example.**

on deep learning technology. In addition to making the target model random errors, an adversarial example can also conduct targeted attacks according to the attacker's wishes and generate specified results. *Eykholt et al. (2018)* show that adversarial examples bring substantial security risks to the application of related technologies. Furthermore, by adding adversarial perturbation to a road sign, the intelligent system may recognize the deceleration sign as an acceleration sign, which will bring substantial hidden dangers to traffic safety.

Currently, the reasons for the adversarial examples are still controversial. *Szegedy et al. (2013)* believed that it is caused by the nonlinearity of the model, while *Kurakin, Goodfellow & Bengio (2016)* propose that the high-dimensional space's linearity is sufficient to generate adversarial examples. If the input samples have sufficiently large dimensions for linear models, they are also attacked by adversarial examples. Adversarial attacks can be divided into single-step attacks, which perform only one step of gradient calculation, such as the FGSM (*Goodfellow, Shlens & Szegedy, 2015*), and iterative attacks, which perform multiple steps to obtain better adversarial examples, such as BIM (*Ren et al., 2020*) or CW (*Carlini & Wagner, 2017*). At the same time, adversarial example attacks can be categorized into white-box, gray-box, and black-box attacks based on the attacker's knowledge. A white-box attack means that the attacker knows all the information, including models, parameters, and training data. We can use it to calculate the attack distance and generate adversarial examples. A gray-box attack means that the attacker knows limited target model information. A black-box attack means that an attacker uses a similar model to generate adversarial examples. The generated adversarial examples have a certain degree of transferability, which can carry out transfer attacks on the model without knowing the relevant information of the model, and it has a high success rate.

Furthermore, extreme samples can even deceive multiple different models. Generally, adversarial examples not only exist in images, but also in speech and text (*Xu et al., 2020*), which make the application of deep learning technology have huge uncertainty and diversity, and there are potential threats at the same time. Therefore, it is urgent to defend against them, which makes the application of deep learning technology have huge uncertainty and diversity, as well as many potential threats.

With the endless emergence of attack methods, the defense of adversarial examples has become a significant challenge. Many defense methods (*Dong et al., 2018*; *Zhang &*

*Wang, 2019*; *Hameed, György & Gündüz, 2020*; *Singla & Feizi, 2020*; *Jin et al., 2021*) have been proposed, such as adversarial training (*Goodfellow, Shlens & Szegedy, 2015*), which increases model robustness by adding adversarial examples to the training process. Some other defenses mainly rely on preprocessing methods to detect or transform the input image before the target network without modifying the target model. For example, *Xu, Evans & Qi (2017)* proposed that the input's adversarial perturbation can be eliminated by reducing the color bit depth of each pixel and spatial smoothing, and they create a defense framework to detect adversarial examples in the input. *Jia et al. (2019)* introduced the ComDefend defense model, which constructs two deep convolutional neural networks: the one for compressing images and retaining valid information; the other for reconstructing images. However, this method does not perform well under the attack of BIM.

On the other hand, if you only perform detection without other measures when defending against adversarial examples, it will not be able to meet actual needs. For example, in an autonomous driving application scenario, the defense system recognizes a road sign and detects that it is an adversarial example. At this time, the defense system refuses to input the image, which will seriously affect its normal operation. In addition, convolutional neural networks are used to extract image features and compress images. If the compression rate is too low, the uncorrupted adversarial perturbation in the reconstruction network will continue to expand, thereby significantly reducing the classifier's accuracy.

To solve the above problems, we propose a defense framework based on image compression reconstruction, which is a preprocessing method. Figure 2 clearly describes the defense framework of this paper. The defense model in the figure can be divided into two steps. The specific operation is to eliminate adversarial perturbations by compressing images to defend against adversarial example attacks. Simultaneously, to ensure that the standard and processed samples do not suffer from performance loss on the target model, we use the deep convolutional neural network to repair the processed images. In short, this paper makes the following contributions:

- To defend against various adversarial example attacks, we propose a defense framework based on image compression and reconstruction with super-resolution. This framework eliminates adversarial perturbations by compressing the input samples and then reconstructs the compressed images using super-resolution methods to alleviate the performance degradation caused by compression.
- As a preprocessing method, there is no need to modify the target model during the defense process, *i.e.,* our method has good performance for single-step and iterative attacks and has a small calculation compared with other adversarial training methods. In addition, it can be combined with different target models to have a protective effect still.
- To verify the effectiveness, applicability, and transferability of the method, extensive experiments of defense tests are carried out on three real data sets and multiple attack methods. The results show that our approach can achieve better defense performance for different adversarial example attacks and significantly reduce image loss.

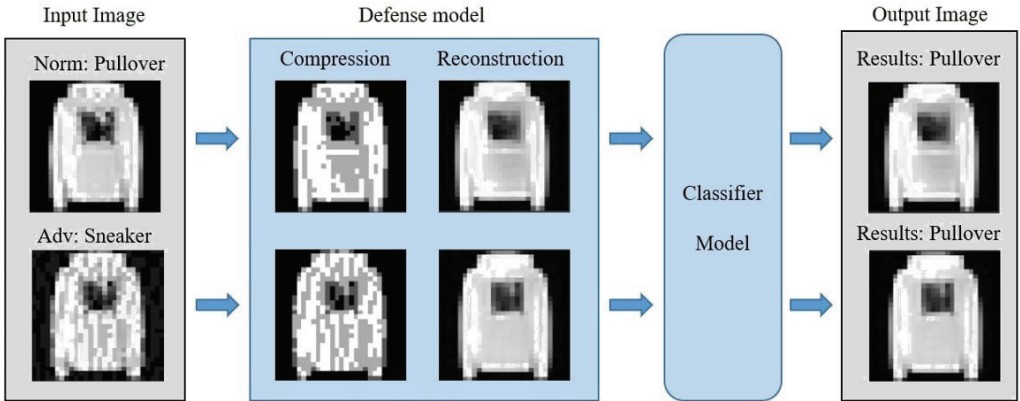

**Figure 2** The defense framework uses input samples as pictures.

The rest of this paper is organized as follows: 'Background' briefly introduces an background of the existing attack and defense methods. 'Our Approach' discusses the methodology and defense framework proposed in this paper in detail, followed by many experiments to demonstrate the feasibility of this method in 'Experiment'. Finally, the conclusion is given in 'Conclusion'.

# BACKGROUND

In this section, we review related works from two aspects: the attack methods of generating adversarial examples and the defensive techniques of resisting adversarial examples.

## Attack methods

In order to verify the versatility of the proposed method, the following four different methods are mainly used to generate adversarial examples.

### Fast gradient sign method (FGSM)

*Goodfellow, Shlens & Szegedy (2015)* proposed the FGSM, a fast and straightforward method of generating adversarial examples. Given the input image, the maximum direction of gradient change of the deep learning model is found, and adversarial perturbations are added to maximize the cost subject to a $L_\infty$ constraint, resulting in the wrong classification result. The FGSM adds the imperceptible perturbations to the image by increasing the image classifier loss. The generated adversarial example is formulated as follows:

$$x^{adv} = x + \varepsilon \cdot sign(\nabla_x J(\theta, x, y_{true})) \tag{1}$$

where $J(\theta, x, y)$ denotes the cross entropy cost function, $x$ is the input image, $y$ is the true label of the input image, and $\varepsilon$ is the hyperparameter that determines the magnitude of the perturbations.

### Basic iterative method (BIM)

The problem of adversarial examples is constantly being studied. *Kurakin, Goodfellow & Bengio (2016)* presented a more direct basic iterative method(BIM) to improve the

performance of FGSM. In other words, BIM is an iterative version of FGSM. It uses the basic idea of gradient descent to perform iterative training with small steps. Moreover, clip the pixel values of the intermediate results after each step to ensure that they are in an $\varepsilon$-neighborhood of the original image:

$$x_0^{adv} = x, \ldots, x_{N+1}^{adv} = clip_{x,\varepsilon}\{x_N^{adv} + \alpha \cdot sign(\nabla_x J(\theta, x, y_{true}))\} \tag{2}$$

Among them, $x$ is the input image, $y_{true}$ is the true class label, $J(\theta, x, y)$ is the loss function, and $\alpha$ is the step size, usually $\alpha = 1$.

This method attempts to increase the loss value of the correct classification and does not indicate which type of wrong class label the model should choose. Therefore, it is suitable for data sets with fewer and different types of applications.

### Carlini & Wagner (C&W)

*Carlini & Wagner (2017)* proposed an optimization-based attack method called *C&W*. *C&W* can be a targeted attack or an untargeted attack. The distortion caused by the attack is measured by three metrics: $(L_0, L_2, L_\infty)$. There are three methods introduced by *C&W*, which are more efficient than all previously-known methods in terms of achieving the attack success rate with the smallest amount of imperceptible perturbation. A successful *C&W* attack usually needs to meet two conditions. First, the difference between the adversarial examples and the corresponding clean samples should be as slight as possible. Second, the adversarial examples should make the model classification error rate as high as possible. The details are shown in Eq. (3).

$$\min \| \frac{1}{2}(\tanh(x_n + 1) - X_n)\|_2^2 + c \cdot f(\frac{1}{2}\tanh(x_n) + 1)$$
$$Where\ f(x^{'}) = \max(\max\{Z(x^{'})_i : i \neq t\} - Z(x_t^{'}), -k) \tag{3}$$

where the $Z$ is the softmax function, the $k$ is a constant used to control the confidence, the $t$ is the target label of misclassification, and $c$ is constant chosen with binary search. In the above formula, $\tanh(x)$ refers to the mapping of adversarial examples to tanh space. After transformation, $x$ belongs to $(-inf, +inf)$, which is more conducive to optimization.

### DeepFool

The DeepFool algorithm is proposed by *Moosavi-Dezfooli, Fawzi & Frossard (2016)*, which generates an adversarial perturbation of the minimum norm of the input sample through iterative calculation. In each iteration, the DeepFool algorithm interferes with the image through a small vector. It gradually pushes the images located within the classification boundary to outside the decision boundary until a misclassification occurs. In addition, DeepFool aggregates the perturbations added in each iteration to calculate the total perturbations. Its perturbations are minor than FGSM, and at the same time, the classifier has a higher rate of misjudgment.

## Defense methods

At present, the defense is mainly divided into two aspects: improving the classifier model's robustness and preprocessing the input without changing the classifier model. Adversarial

training (*Goodfellow, Shlens & Szegedy, 2015*) is currently a more effective defense method proposed by Goodfellow et al. They use adversarial examples to expand the training set and train with the original samples to increase the model's fit to the adversarial examples, thereby improving the robustness of the model. However, this increases the calculation cost and complexity, and adversarial training has excellent limitations. When facing adversarial attacks generated by different methods, the performance varies significantly.

Generally, the preprocessing process does not need to modify the target model, compared with adversarial training and other methods, which is more convenient to implement. Moreover, it has a smaller amount of calculation and can be used in combination with different models. For instance, *Xie et al. (2017)* propose to enlarge and fill the input image randomly. The entire defense process does not need to be retrained and is easy to use. However, the results show that this method is only effective for iterative attacks such as C&W and DeepFool (*Moosavi-Dezfooli, Fawzi & Frossard, 2016*), while for FGSM, the defensive effect of this single-step attack is inferior. They believe that this is due to the iterative attack to fitting the target model, resulting in low-level image transformation that can destroy the fixed structure of the adversarial disturbance. In addition, *Liao et al. (2018)* regard adversarial perturbation as a kind of noise, and they design a high-level representation guided denoiser (HGD) model to eliminate the adversarial disturbance of the input species. *Das et al. (2017)* used JPEG compression to destroy adversarial examples.

Similarly, Pixel Defend (*Song et al., 2017*) is a new method that purifies the image by moving the maliciously perturbed image back to the training data to view the distribution. Feature squeezing (*Xu, Evans & Qi, 2017*) is both attack-agnostic and model-agnostic. It can reduce the image range from [0, 255] to a smaller value, merge the samples corresponding to many different feature vectors in the initial space, and reduce the search space available to the opponent. Similar methods also include label smoothing (*Warde-Farley & Goodfellow, 2016*), which converts one-hot labels to soft targets. Besides, *Zhang et al. (2021)* proposed a domain adaptation method, which gradually aligns the features extracted from the adversarial example domain with the clean domain features, making DNN more robust and less susceptible to spoofing by diverse adversarial examples.

## OUR APPROACH

### Motivation

The essence of adversarial examples is to deliberately add high-frequency perturbations to clean input samples and amplify the noise through deep neural networks so that the model gives the wrong output with high confidence. For example, when we input a clean image of a cat, add an imperceptible perturbation, the classifier will misclassify it as a leopard with high confidence. Through previous research, we have also learned that the classifier is robust to ordinary noise. Simultaneously, the adversarial perturbation in the adversarial example is very unstable and can be destroyed by some simple image transformation methods. According to the currently known image characteristics, we use image processing methods to eliminate the fixed structure of the adversarial perturbation before the adversarial example is input to the target. At the same time, to ensure the system's normal operation

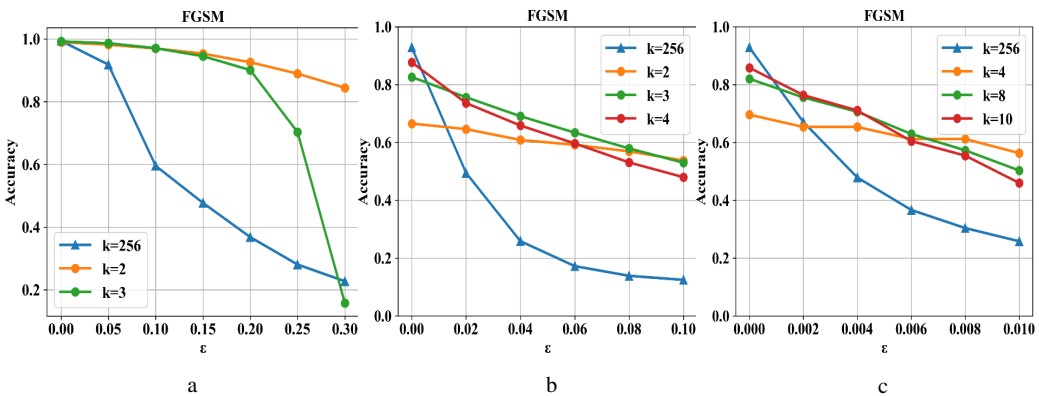

**Figure 3** Changes in the defense effect of pixel compression on the three data sets.

and the performance of the adversarial examples after converting the original image and the target model, we combine image compression and image restoration neural networks to form the entire defense model. This model can convert adversarial examples into clean images to resist adversarial example attacks without significantly reducing the quality of ordinary images.

## Pixel depth reduce

An array of pixels represents a standard digital image in a computer, and each pixel is usually represented as a number with a specific color. Since two common representations are used in the test data set, they are 8-bit grayscale and 24-bit color. Grayscale images provide $2^8 = 256$ possible values for each pixel; we use $k$ to represent the maximum range of pixel values. An 8-bit value represents a pixel's intensity, where 0 is black, 255 is white, and the average number represents different shades of gray. The 8-bit ratio can be expanded to display color images with separate red, green and blue channels and provides 24 bits for each pixel, representing $2^{24} \approx 16$ million different colors. The redundancy of the image itself offers many opportunities for attackers to create adversarial examples.

Compressed pixel bit depth can reduce image redundancy and destroy the fixed structure of adversarial examples in the input while retaining image information without affecting the image's accuracy on the classifier model. As shown in Fig. 3, the defense capability is tested on the MNIST, Fashion-MNIST, and CIFAR-10 datasets. In the sub-pictures (a), (b), and (c), $k$ refers to the maximum range of pixel value color depth. When $\varepsilon$ is small, the attack intensity is low, and reducing each pixel's color depth can have an excellent defense effect. On the contrary, as the attack intensity continues to increase, the defense effect is also declining. At the same time, the situation becomes more complicated in the face of more complex data sets (such as Cifar-10). Although a higher compression rate can improve the defensive performance to a certain extent, it will also cause the loss of ordinary image information and reduce the prediction accuracy of the classifier model. Therefore, we need to repair the damaged image after compression.

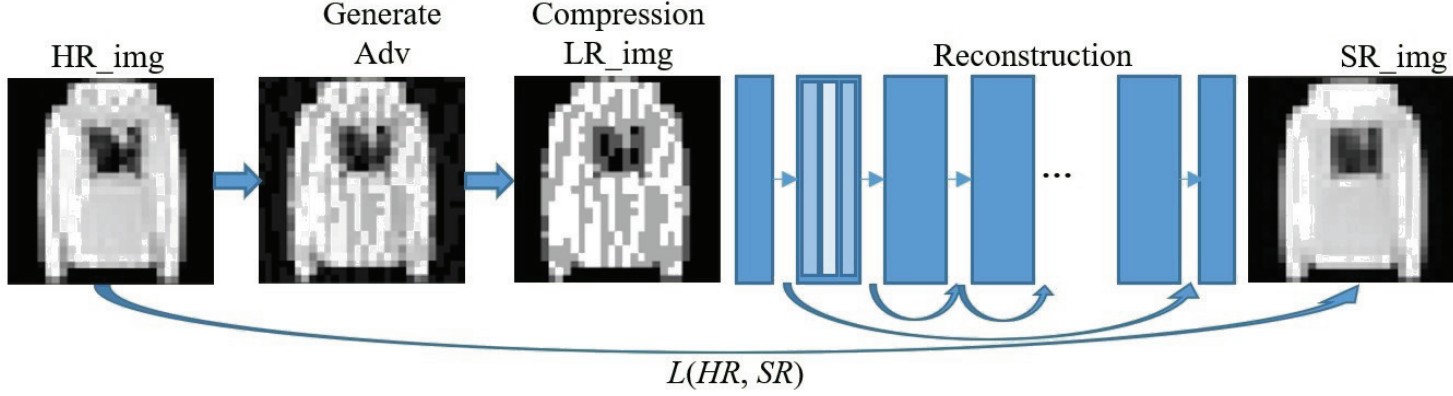

**Figure 4** Implementation framework of our defense model.

## Image reconstruction

Super-resolution (*Mustafa et al., 2019*) image reconstruction is a typical application in computer vision and deep learning development. Recently, super-resolution has made significant progress, and this technology is often used to reconstruct high-resolution images or repair damaged images. We also hope to use this deep neural network to repair compressed images. Generally, the low-quality images used in the training process of the image reconstruction network are obtained through down-sampling, blurring, or other degradation methods. In this paper, we first collect low-quality images by compressing the pixel depth and input them into the reconstruction network; then, we train the deep neural network learning ability to restore low-quality images to high-quality images.

Without loss of generality, the deeper the network and the more parameters, the better the performance for deep convolutional neural networks. Figure 4 shows the main process of applying the input image to reconstruct the defense model based on image compression. For reconstructing network structure, we refer to the excellent EDSR structure in the super-resolution image reconstruction network (*Lim et al., 2017*) to build a very deep neural network to ensure the recovery performance of the image. The chain-hopping structure in the network (*He et al., 2016*) can help us build a deeper network to obtain better performance without worrying about gradients' disappearance. The entire defense model is used to complete the conversion from adversarial examples to clean samples and ensure the quality of the reconstructed image. Without considering the disappearance of the gradient, to build a deeper network, we add the ResNet structure to the reconstruction network, and use the ReLU activation function and a $3 \times 3$ filter. In the reconstruction training process, we first train the low-multiple up-sampling model and then initialize the high-multiple up-sampling model with the parameters obtained in training. This will make the training time of the high-multiple upsampling model shorter and the training result better. Finally, we get a picture *SR_img* that eliminates the perturbation of adversarial examples.

In the experiment, we find that after compressing the high-strength adversarial example, the classification result is different from the original image and the adversarial

example. Because the high compression rate destroys the fixed structure of the adversarial perturbation, it also causes a certain amount of information loss. Adding adversarial examples in the training process can solve this problem well and improve the neural network's ability to repair compressed images. We use clean samples to generate adversarial examples during the training process and keep the total number of training sets unchanged. Then we use adversarial samples for training and use clean data sets as labels to narrow the gap between adversarial samples and clean samples. To prevent the network from overfitting and repair the compressed image of the adversarial example, we reduce the repair effect of the ordinary sample after compression to a certain extent.

To better reconstruct clean samples, we minimize the distance between the reconstructed *SR_img* and the original image *HR_img*. We use Mean Squared Error(MSE) to define the loss function of the CNN:

$$L(\theta) = \frac{1}{2N} \sum \|F(X, \theta) - Y\|^2 \tag{4}$$

where $F$ is the image restoration network, $X$ is the compressed image, $\theta$ is the network parameter, and $Y$ is the original image.

After the training is completed, the reconstructed network has the ability to filter and fight noise. We add the reconstructed network model before the classifier that needs to be protected. When a batch of samples are input, they first pass through our reconstructed network model. If these input images include adversarial examples, their adversarial features will be destroyed, while normal samples will not be affected. In this way, we can turn the input into a clean sample to defend against adversarial attacks.

## EXPERIMENT

In this section, we use experiments to verify the effectiveness of the proposed algorithm. The basic process of the experiment includes generating adversarial examples on different datasets and training multiple classifier models to test the performance and transferability of the defense model. In addition, we conduct a comprehensive theoretical analysis of the experimental results.

### Experimental setup

In our experiments, we use three different image datasets: MNIST (*LeCun et al., 1998*), Fashion-MNIST (F-MNIST) (*Xiao, Rasul & Vollgraf, 2017*) and CIFAR-10 (*Xiao et al., 2018*). The MNIST and F-MNIST datasets both contain 60,000 training images and 10,000 test images. Each example is a 28 × 28 grayscale image associated with one label in 10 categories. The difference is that MNIST is a classification of handwritten numbers 0–9, while F-MNIST is no longer an abstract symbol but a more concrete clothing classification. The CIFAR-10 dataset is a 32 × 32 color image associated with 10 category labels, including 50,000 training images and 10,000 test images. To prevent over-fitting, both the defense model and the classifier target model in this paper are trained by the training set. The classifier model's accuracy and the defense model's performance experiment are conducted in the test set.

To verify the generalization effect of the defense framework, this paper chooses FGSM, BIM, DeepFool and *C&W* four methods to generate different types of adversarial examples for defense testing. We preprocess the defense model and then input it into the classifier model to get the experimental results. For FGSM and BIM, we use the $L_\infty$ norm to control the perturbation's intensity by changing the size of $\varepsilon$. Differently, we use the $L_2$ norm to implement the *C&W* model, and adjust the degree of perturbation by controlling the maximum number of iterations. To preserve the original image information and eliminate adversarial perturbations as much as possible, we set $k = 2$ ($k$ denotes the maximum range of pixel value color depths) on the MNIST dataset and $k = 4$ on the F-MNIST dataset.

## Experiment results

In this section, the adversarial examples generated by FGSM, BIM, DeepFool, and *C&W* on different datasets are applied to the defense framework of this paper. Simultaneously, in the training process, to make the reconstructed network have selective noise reduction and generalization capabilities, we use the FGSM with the most perturbation to generate adversarial examples and input them into the neural network. Generally, simple images need to add a significant perturbation to be effective. In this paper, for the MNIST dataset, the value of $\varepsilon$ is up to 0.3; for the F-MNIST dataset, the value of $\varepsilon$ is from 0 to 0.1; for the Cifar-10 dataset, the value of $\varepsilon$ is taken from 0 to 0.01. When $\varepsilon$ is equal to 0.01, the adversarial example is enough to produce a higher error rate on the target classifier model for the CIFAR-10 data set. The results of each step of the defense experiment process are shown in the figure below.

From left to right, the different subgraphs in Fig. 5 are the adversarial examples generated by FGSM, BIM, DeepFool, and *C&W* attack methods, respectively; from top to bottom are normal examples, adversarial examples, compression examples, and reconstructed examples. Figure 5A is the result of working on the MNIST data set. It can be seen that only the pixel compression operation can eliminate most of the adversarial perturbations, and the adversarial examples restore the accuracy of the classifier model. In addition, the adversarial examples generated by different methods have different perturbation levels to the image, and FGSM has the most massive perturbation level. When $\varepsilon$ is 1.5, it has already had a more significant impact on the image, and the human eye can already detect it, but our method can still restore it to a clean sample. A few extreme adversarial examples become other classification results after processing, as shown in the first column of Fig. 5A. Still, after the reconstruction of the network, the recognition accuracy is also restored.

Figures 5B and 5C show the experimental results of the relatively complex of F-MNIST and CIFAR-10 data sets. Since pixel depth reduction is a lossy compression, choosing an appropriate compression level can eliminate the adversarial perturbation of the input sample as much as possible while retaining the necessary information. Generally, a slight loss of details does not affect the classifier model's correct recognition of the image. The following experiment will specifically show the defense effect of different data sets after processed by our defense framework under different attack intensities.

Figures 6A–6D are the recognition accuracy rates of the model ResNet-50 with and without defensive measures for different attack strengths ($\varepsilon$, iteration) on the MNIST
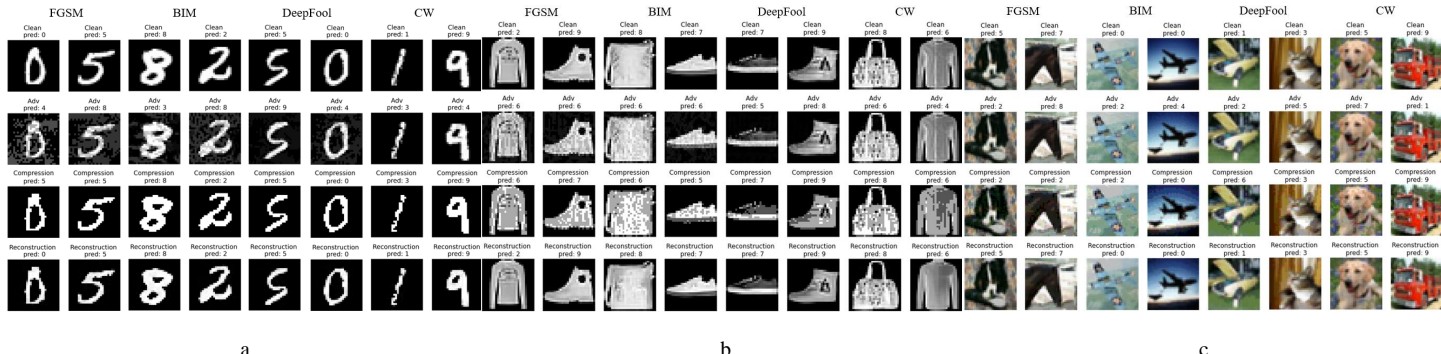

**Figure 5   Performance of defense models against multiple adversarial example attacks on different datasets.**

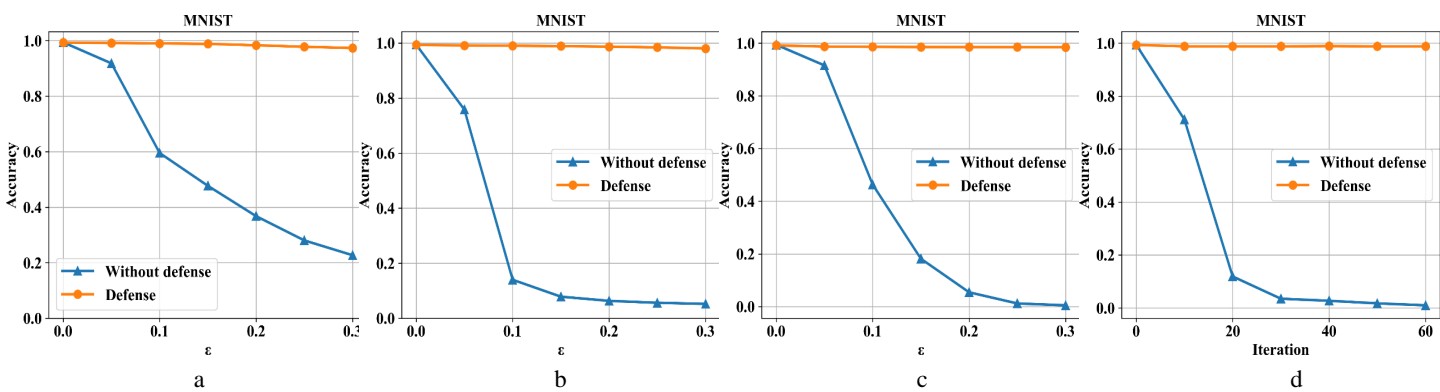

**Figure 6   Performance of the defense model on the MNIST dataset.**

dataset. Our algorithm is compared with four different types of adversarial samples in defensive and non-defensive situations. After the defense model processes the data set in this paper, the accuracy of the original image has almost no change. Furthermore, in the face of different types of attacks from FGSM, BIM, DeepFool, and $C\&W$, the operation can eliminate adversarial perturbations in the input image. This is because the defense model has certain image recovery capabilities, the MNIST image structure is relatively simple, and the information is not easily damaged. For FGSM attacks, we can see that the accuracy can be restored from 20% to 97% under high attack intensity, BIM can be restored from 5% to 98%, DeepFool can be restored from 0% to 98%, and $C\&W$ can be restored from 0% to 98%.

Figures 7A–7D are the recognition accuracy rates of the model ResNet-50 with and without defensive measures for different attack strengths ($\varepsilon$, iteration) on the Fashion-MNIST dataset. It can be seen from Fig. 7 that we have also achieved good results in the face of a slightly complicated Fashion-MNIST defense model, i. e., the original image recognition accuracy rate drops by 4% after preprocessing. For FGSM attacks, it recovery
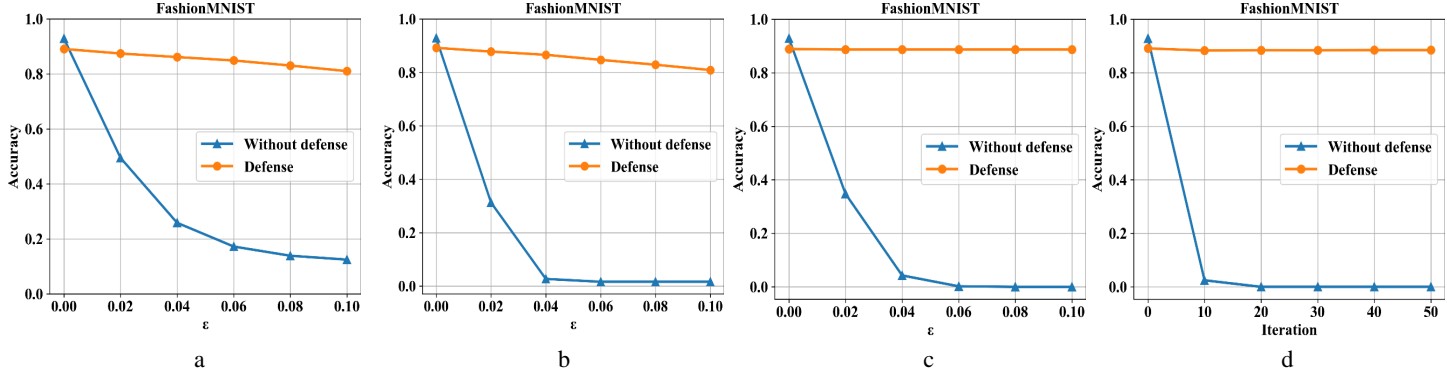

**Figure 7  Performance of the defense model on the F-MNIST dataset.**

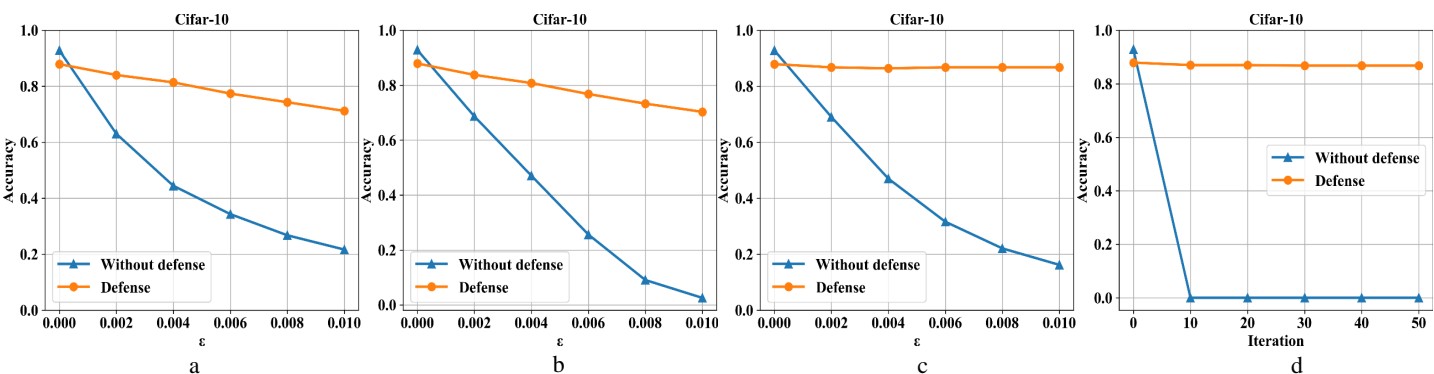

**Figure 8  Performance of the defense model on the CIFAR-10 dataset.**

from 13% to 81%; for BIM, it recovery from 1% to 82%; for DeepFool, it recovery from 0% to 88%; and for *C&W*, it recovery from 0% to 88%.

Figures 8A–8D are the recognition accuracy rates of the model ResNet-50 with and without defensive measures for different attack strengths ($\varepsilon$, iteration) on the Cifar-10 dataset. When processing the three-channel color dataset CIFAR-10, we find that it is more complicated than the first two single-channel grayscale image data sets. Mainly because it is difficult to balance the pixel compression rate and the defense rate, which makes the defense effect appear to be reduced to a certain extent. It can be seen from Fig. 8, the ordinary sample has a loss close to 5% in accuracy after compressed and reconstructed. For FGSM attacks, the defense model can restore the accuracy from 23% to 71%, BIM from 2% to 70%, DeepFool from 18% to 87%, and CW from 0% to 87%.

## Defense transferability

As a preprocessing method, we can combine different target models without modifying them. To verify the defense model's portability, we train three classifier models from weak to strong performance. They are: LeNet (*LeCun et al., 1998*), GoogLeNet (*Szegedy et al.,*

**Table 1  The performance of the defense model combined with LeNet and GoogLeNet on MNIST.**

| Dataset | Network | Clean | FGSM | BIM | DeepFool | CW |
|---|---|---|---|---|---|---|
| | ResNet50 (no defense) | 99% | 48% | 9% | 18% | 3% |
| | ResNet50 (defense) | 99% | 98% | 99% | 99% | 99% |
| MNIST | LeNet (no defense) | 99% | 38% | 2% | 9% | 3% |
| ($\varepsilon = 0.15$) | LeNet (defense) | 99% | 98% | 98% | 98% | 98% |
| | GoogLeNet (no defense) | 99% | 48% | 26% | 19% | 15% |
| | GoogLeNet (defense) | 99% | 98% | 97% | 98% | 98% |

**Table 2  The performance of the defense model combined with ResNet and GoogLeNet on F-MNIST.**

| Dataset | Network | Clean | FGSM | BIM | DeepFool | CW |
|---|---|---|---|---|---|---|
| | ResNet50 (no defense) | 93% | 22% | 2% | 0% | 0% |
| | ResNet50 (defense) | 89% | 85% | 85% | 89% | 89% |
| F-MNIST | GoogLeNet (no defense) | 90% | 35% | 18% | 2% | 0% |
| ($\varepsilon = 0.05$) | GoogLeNet (defense) | 90% | 81% | 84% | 88% | 88% |
| | ResNet101 (no defense) | 93% | 20% | 2% | 0% | 0% |
| | ResNet101 (defense) | 89% | 83% | 84% | 88% | 88% |

**Table 3  The performance of the defense model combined with ResNet and GoogLeNet on Cifar-10.**

| Dataset | Network | Clean | FGSM | BIM | DeepFool | CW |
|---|---|---|---|---|---|---|
| | ResNet50 (no defense) | 84% | 64% | 23% | 20% | 39% |
| | ResNet50 (defense) | 79% | 68% | 59% | 72% | 72% |
| CIFAR-10 | GoogLeNet (no defense) | 98% | 36% | 34% | 35% | 0% |
| ($\varepsilon = 0.005$) | GoogLeNet (defense) | 94% | 51% | 52% | 61% | 60% |
| | ResNet101 (no defense) | 84% | 64% | 24% | 22% | 43% |
| | ResNet101 (defense) | 80% | 69% | 63% | 74% | 74% |

*2015*), and ResNet101 (*He et al., 2016*). Besides, we combine the defense model trained with these three classifier models to test the defense performance.

Tables 1 and 2 show in detail the experimental results of the transferability of the defense model. On the MNIST and Fashion-MNIST datasets, we take the median value of 0.15 and 0.05 for $\varepsilon$, respectively. Due to the performance difference of the target model, the effect will be slightly reduced when the defense model is combined with different models. However, it can still defend well against adversarial example attacks. Table 3 shows the transferability performance of our defense model combined with ResNet 50, ResNet101, and GoogLeNet on the data set Cifar-10. We take the median value of 0.005 for $\varepsilon$. The classification accuracy of the overall network model on the Cifar-10 data set has been reduced compared to the performance of the MNIST and F-MNIST data sets. This is because the Cifar-10 data set is relatively complex. In short, the classification accuracy of the network model with defense is much higher than the network model without defense when facing different attacks.

**Table 4  The result of comparisons with other defensive methods (F-MNIST).**

| Network | Methods | Clean | FGSM | BIM | DeepFool | CW |
|---|---|---|---|---|---|---|
| | Normal | **93%/93%** | 38%/24% | 00%/00% | 06%/06% | 00%/00% |
| | Adversarial FGSM | **93%/93%** | 85%/85% | 51%/00% | 63%/07% | 67%/21% |
| | Adversarial BIM | 92%/91% | 84%/79% | 76%/63% | 82%/72% | 81%/70% |
| Resnet50 | Feature squeezing | 84%/84% | 70%/28% | 56%/25% | 82%/72% | 83%/83% |
| | Pixel defend | 89%/89% | 87%/82% | 85%/83% | 83%/83% | 88%/88% |
| | ComDefend | **93%/93%** | **89%/89%** | 70%/60% | 88%/88% | 88%/89% |
| | Our method | 89%/89% | 87%/86% | **87%/86%** | **90%/89%** | **89%/89%** |

## Performance comparison between similar defense models

This section uses four methods (FGSM, BIM, DeepFool, and *C&W*) on the Fashion-MNIST
dataset to generate two adversarial examples of different strengths for the ResNet50 target
model and conduct defense tests. To better compare with other classic methods and verify
the effectiveness of our approach, all experiments use the same dataset, target model, and
related parameter settings as other methods. As shown in Table 4, our method performs
best compared with other methods under attack models such as BIM, DeepFool, and
*C&W*.

   Although the ComDefend method is better at preserving the original image information,
it adds Gaussian noise during training to improve the network's ability to resist noise. The
defense effect of some attacks, such as BIM, is not ideal. The impact of adding an FGSM
attack is only acceptable in the case of FGSM adversarial examples, and it performs poorly
for adversarial examples generated by other methods. In general, although the direct
pixel depth reduction has made a certain sacrifice in image information preservation, the
confrontation samples generated in the face of different attacks in the above experiments
can all play a good defense effect. Therefore, to the best of our knowledge, our method can
effectively defend against adversarial example attacks.

## CONCLUSION

Finding a robust defense method for adversarial examples is an open problem, and many
researchers have carried out work in this area. This paper proposes an image compression
and reconstruction defense framework to defend against adversarial example attacks based
on the redundancy of images and the sensitivity of adversarial examples. We compress the
pixel bit depth in the image to destroy the adversarial perturbation of the image and then
use DNN to repair the image. On the premise of ensuring the image quality, the adversarial
examples are converted into clean samples to achieve the purpose of defense. In addition,
this method can be easily combined with other defense methods without modifying the
target classifier model. Extensive experiments have been applied to the three real datasets
of MNIST, F-MNIST, and CIFAR-10, showing the superiority of the proposed method
over some classic techniques to defend against adversarial examples, *i.e.,* the defensive
framework we designed can resist different attacks. However, due to limited knowledge
and personal abilities, many issues need further research. We will study how to better

balance the compression rate of complex images and preserve adequate information and verify the method's effectiveness on more complex datasets.

### Funding

This work was supported by the National Natural Science Foundation of China under Grant 61572034, in part by the Major Science and Technology Projects in Anhui Province under Grant 18030901025, and by the Natural Science Foundation of Anhui Province of China under Grant 2008085MF220. There was no additional external funding received for this study. The funders had no role in study design, data collection and analysis, decision to publish, or preparation of the manuscript.

### Grant Disclosures

The following grant information was disclosed by the authors:
The National Natural Science Foundation of China: 61572034.
The Major Science and Technology Projects in Anhui Province: 18030901025.
The Natural Science Foundation of Anhui Province of China: 2008085MF220.

### Competing Interests

The authors declare there are no competing interests.

### Author Contributions

- Yu(AUST) Zhang conceived and designed the experiments, performed the experiments, analyzed the data, performed the computation work, prepared figures and/or tables, authored or reviewed drafts of the paper, and approved the final draft.
- Huan Xu conceived and designed the experiments, analyzed the data, performed the computation work, prepared figures and/or tables, authored or reviewed drafts of the paper, and approved the final draft.
- Chengfei Pei conceived and designed the experiments, performed the experiments, performed the computation work, prepared figures and/or tables, authored or reviewed drafts of the paper, and approved the final draft.
- Gaoming Yang conceived and designed the experiments, performed the experiments, analyzed the data, authored or reviewed drafts of the paper, and approved the final draft.

### Data Availability

The source code is available in the Supplemental Files and at GitHub: https://github.com/yuzhang866/DEFENSE_ADV.

### Supplemental Information

Supplemental information for this article can be found online at http://dx.doi.org/10.7717/peerj-cs.811#supplemental-information.

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
