# Peer review of "Adversarial example defense based on image reconstruction"

_PeerJ Computer Science, doi:10.7717/peerj-cs.811_

## Round 0.1 · original submission · Major Revisions

The authors should address all the proposed comments and improvements that reviewers highlight. There are many comments that the authors should revise accordingly.

Reviewer 1 ·

Basic reporting

1. The paper is easy to understand, but the English language should be improved on grammar and tense. Some examples include lines 241-242, 273, 288-289, and the inconsistency of tense used in Section experiment results.

2. The authors missed some background information on adversarial attacks and defense methods. In the experiment, you used DeepFool attack and compared your proposed method with Pixel Defend, Feature Squeezing, and ComDefend but you didn't introduce these methods in Section Background.

3. The structure of this paper is good. Figures are relevant to the content of the paper. Some problems are in Table 3. The names of the first two columns should be "network" and "method". The caption of Table 3 doesn't provide sufficient information to help understand this table. Specifically, why for each cell there are two numbers (e.g., 93%/93%)? I also don't find the explanation in the paper.

4. I thank you for providing the source code, but you need also to provide a Readme file to describe how to use the code. In addition, there are many Chinese characters in the code and filenames, which should be replaced with English words. The authors should also provide pre-trained models to ensure the experimental results are reproducible.

Experimental design

1. The paper proposed an adversarial defense method based on image compression and reconstruction. However, the detailed structure of the image reconstruction network was not introduced. In the provided source code I saw some different network architectures used on different datasets. The authors should introduce them in the paper.

2. In lines 214-216, the authors described they generated adversarial examples and used them with clear examples to train the image reconstruction network. However, in the source code (cs-59969-MNIST-002.zip/train_turn_defense.ipynb), I found only adversarial examples were used. The authors should double-check the used method and describe it clearly and accurately.

3. The authors conducted experiments on evaluating the performances of the proposed method, the transferability of the proposed methods, and the comparison of the proposed method with other defense methods. The research questions are well defined and meaningful. However, the experiment setting should be improved in some ways.

First, ImageNet or tiny-imagenet dataset should be used to evaluate the proposed method because it is the most widely used and most complicated dataset for image classification.

Second, for the comparison of the proposed method and other defense methods, more similar defense methods based on image reconstruction such as HGD mentioned in the paper could be considered to compare.

Validity of the findings

1. For the experiment of comparing the proposed method with other defense methods, only result on F-MNIST was reported. The results on the other two datasets should be provided to prove the proposed method is better than other methods in general.

Reviewer 2 ·

Basic reporting

The introduction section demands to be more convincing. Try to structure the introduction section with four paragraphs as follows: i) State the motivation and clearly define the problem to be solved. ii) Make a thorough discussion of the state-of-the-art. iii) Describe your proposal in fair context to other published methods highlighting advantages and disadvantages of these methods. iv) Clearly pinpoint the novelty/contribution of your proposal and briefly describe your findings.

Experimental design

The performance of CNN strongly depends on an optimum structure of a network. The training structure in Figure 4 needs to be self-contained such as number of layers, height and width of each layer.

Networks with defense method show degraded performance for clean images in Table 2. How do you validate this result?

Validity of the findings

Network models in supplement files were not possible to test. It is needed to provide comprehensive readme files to run and test source codes, models and dataset.

Is there any limitation of the proposed methodology?

Additional comments

The manuscript is overall well written. If there are weaknesses, as I have noted above which need be improved upon before publication.

---

## Round 0.2 · Minor Revisions

The issues highlighted by Reviewer#1 must be addressed before your manuscript can be published.

Reviewer 1 ·

Basic reporting

1. The paper proposed an adversarial defense method by combining image compression and image reconstruction models. The background of adversarial attack and defense was introduced in detail. The authors conducted comprehensive experiments to evaluate the performance of the proposed method against common adversarial attacks and compared the method with existing defenses. The experiment result shows that the proposed method achieves good performance.

2. There are a few tense inconsistencies in the Section background, which should be revised.

3. In the Section Approach, the overview of the defense method should be described more clearly. The authors could consider showing how the defense method works from taking as the input of the original image to reporting whether the image is an adversarial example in Figure 4. Now Figure 4 just shows the process until the image reconstruction. I cannot easily know how to use the output image to detect adversarial examples from the Figure.

Experimental design

The authors conducted experiments on evaluating the performances of the proposed method, the transferability of the proposed methods, and the comparison of the proposed method with other defense methods. The experiment results show the proposed method outperforms other baseline defenses. In Table 4, the author should highlight (bold) the best experiment results, which is better to help compare the performances of these methods.

Validity of the findings

The experiment results are well evaluated. The authors provided the source code and detailed instructions for reproducing the experiment.

Reviewer 2 ·

Basic reporting

The revised manuscript has been improved and addressed all the concerns accordingly. The usage of English language is satisfactory.

Experimental design

The experiential design and analysis is satisfactory in the revised manuscript.

Validity of the findings

no comment

---

## Round 0.3 · accepted · Accept

Based on the reviewer's comments, I'm glad to tell you that your paper has been accepted for publication.

Reviewer 1 ·

Basic reporting

The revised manuscript has addressed my questions accordingly.

Experimental design

Experiments are solid in the revised manuscript.

Validity of the findings

no comments